# Vehicle Detection in High-Resolution Aerial Images with Parallel RPN and Density-Assigner

**Xianghui Kong** [1], **Yan Zhang** [1], **Shangtan Tu** [2], **Chang Xu** [1] and **Wen Yang** [1,*]

[1] School of Electronic Information, Wuhan University, Wuhan 430072, China; martinkong@whu.edu.cn (X.K.)
[2] Shanghai Institute of Satellite Engineering, Shanghai 201109, China
* Correspondence: yangwen@whu.edu.cn; Tel./Fax: +86-27-6875-4367

**Abstract:** Vehicle detection in aerial images plays a significant role in many remote sensing applications such as city planning, road construction, and traffic control. However, detecting vehicles in aerial images remains challenging due to the existence of tiny objects, the scale variance within the same type of vehicle objects, and dense arrangement in some scenarios, such as parking lots. At present, many state-of-the-art object detectors cannot generate satisfactory results on vehicle detection in aerial images. The receptive field of the current detector is not fine enough to handle the slight scale variance. Moreover, the densely arranged vehicles will introduce ambiguous positive samples in label assignment and false predictions that cannot be deleted by NMS. To this end, we propose a two-stage framework for vehicle detection that better leverages the prior attribution knowledge of vehicles in aerial images. First of all, we design a Parallel RPN that exploits convolutional layers of different receptive fields to alleviate the scale variation problem. To tackle the densely arranged vehicles, we introduce a density-based sample assigner in the vehicle-intensive areas to reduce low-quality and occluded positive samples in the training process. In addition, a scale-based NMS is proposed to filter out redundant proposals hierarchically from different levels of the feature pyramid. Moreover, we construct two challenging vehicle detection datasets based on the AI-TOD and xView datasets which contain many tiny objects. Extensive experiments on these two datasets demonstrate the effectiveness of our proposed method.

**Keywords:** vehicle object detection; region proposal network; label assignment; nonmaximum suppression; dataset





## 1. Introduction

Object detection is a classic task of computer vision which simultaneously predicts object category and location. As the global economy continues to grow, the number of vehicles increases a lot, making the task of collecting and analyzing vehicle data particularly important. Compared with ground sensors, remote sensing sensors have a wide vision field so as to obtain vehicle information more efficiently [1]. Therefore, vehicle detection in aerial images has broad applications in city planning, disaster rescue, traffic control, etc.

Due to the particularity of aerial imaging view and long imaging distance [2], optical aerial images contain larger and more complex spatial scenes than natural scenes, resulting in an enormous number of tiny-sized objects, large-scale variance, and the dense arrangement issue in some scenarios such as the parking lot [3]. Currently, some widely-used object detection methods based on deep learning such as Faster R-CNN [4], Cascade R-CNN [5], and CenterNet [6] are designed to detect general objects in daily life. Unfortunately, they often struggle to generate satisfactory results on the aerial vehicle object detection task.

In light of the suboptimal performance of generic object detectors with the vehicle detection task, some methods have been specialized for vehicle detection. For example, researchers have proposed methods to enhance the small or tiny object detection performance in vehicle detection [7,8]. In addition, several works note that vehicles in aerial images are arbitrary-oriented, and, thus, design orientation-aware methods [9–11].

These methods have certainly greatly pushed forward vehicle detection research. However, there are still many issues to be tackled. The scale of vehicles approximates continuous changing, while the receptive field of different feature layers in the commonly adopted feature pyramid network (FPN) [12] changes discretely. Thus, the receptive field is not fine enough to fit all-sized objects. Furthermore, there are many densely arranged vehicles in the aerial imagery, including the congested highway, the parking lot, and vehicles working on construction sites. These densely arranged vehicles notably deteriorate the detection performance of the label assignment and post-processing. Concretely, in the label assignment process, many ambiguous positive training samples are introduced to the dense region, leading to harmful gradients and inaccurate predictions[13] . In the post-processing process, the nonmaximum suppression (NMS) operation, which is locally optimal, fails to properly delete correct redundant predictions in the dense region.

In our work, we tackle the receptive field mismatch issue of vehicles via a newly designed parallel RPN and enhance the detection of densely arranged vehicles by the Density-assigner along with a scale-based NMS. In addition, we contribute two challenging vehicle detection datasets to the community.

The main contributions of our work are as follows.

- We propose a Parallel RPN to generate proposals with high quality by incorporating different convolutional kernels, obtaining finer receptive field change, and thus fitting a broader scale range of objects.
- We design a density-based label assignment strategy in a dividing and conquering manner, which separates the assignment of the dense region and sparse region. We further propose a scale-based NMS strategy to suppress overlapped anchors generated from different FPN output layers.
- We construct two vehicle detection datasets based on the AI-TOD dataset and the xView dataset and verify the effectiveness of the proposed method on them.

The rest of this paper is organized as follows: related works are discussed in Section 2; the details of the proposed method are described in Section 3; experiment results are presented in Section 4; discussion and failure cases regarding our work are in Section 5; and, finally, Section 6 concludes the paper.

## 2. Related Works

Vehicle detection is of great interest in the remote sensing community. Recent methods on the vehicle detection can be separated into the following four aspects: multiscale object detection methods, tiny object detection methods, densely arranged object detection methods, and other specially designed methods.

### 2.1. Multiscale Learning

Image pyramid is a classical image scale transformation method, in which a series of images with different sizes are obtained through upsampling or downsampling of original images to construct different-scale spaces, thus improving the detection performance of vehicles. The feature pyramid network (FPN) [12] builds a top-down structure with horizontal connections to generate a series of scale-variant feature maps, which can improve the performance of small objects without much overhead. On the basis of FPN, Liang et al. proposed DFPN [14], which makes the semantic features of small objects more sensitive to be extracted by the network. Based on SSD [15], Cao et al. proposed a multilevel feature fusion algorithm that introduced contextual content information, namely, Feature Fusion SSD (FFSSD) [16]. Hu et al. [17] proposed that the relationship network can be used to build models between objects by establishing relations between appearance and geometric features. Chen et al. [18] used a context patch that was combined with proposals generated by the region proposal network (RPN) and enhanced the performance of R-CNN.

## 2.2. Tiny Object Detection

A simple and effective method to improve tiny object detection training strategy is to reduce the IoU threshold when assigning positive and negative samples for anchors in RPN [19]. It is easier for ground truths to match anchors. However, some low-quality positive samples are introduced, which will impact the network. Zhang et al. proposed an adaptive training sample selection strategy (ATSS) [20] to automatically select positive and negative samples for network training according to the statistical characteristics of objects. In addition, Singh et al. proposed a scale-normalization method SNIP [21,22], which selectively trains samples within a certain scale range. As a result, SNIP solves the problem of network performance degradation caused by drastic scale changes. Xu et al. [23] designed a normalized Wasserstein distance to select training samples in RPN to generate proposals of high quality, which is the state-of-the-art method in tiny object detection.

## 2.3. Densely Arranged Object Detection

Considering the large scene of the aerial imagery, densely and compactly arranged objects are ubiquitous in it. For example, ships are mooring in the harbor, and aerial planes are in the aircraft cemetery. Similarly, large amounts of vehicles can be found along roads or in the parking lot. To detect the densely arranged objects, some previous works mainly tackle the feature coupling issue. If the feature receptive fields of compactly arranged objects are aliased, they will bring about inaccurate feature learning and deviated bounding box predictions. The attention mechanism is used by SCRDet [24] to suppress background regions and enhance the objects' main body. The CFA [25] presents a convex-hull instance representation for sample learning in oriented object detection. The convex-hull can compactly cover the main body of the instance compared to previous works, greatly eliminating the aliasing issue.

## 2.4. Specialized Vehicle Detectors

In addition, some other methods specialized for vehicle detection in aerial images have been proposed. Ophoff et al. [26] adjusted hyperparameters in four different single-shot object detection networks specialized for vehicles and vessel detection. A new model was proposed by Chen et al., named hybrid deep neural network (HDNN) [27]. The main idea of this model was to eliminate the feature difference of vehicles by replicating the convolutional layers at different scales in order to detect vehicles with significant multiple scales. Although the idea was full of innovation, and the improvement of vehicle detection rate compared to other solutions was substantial at the time, this approach did not tackle the problem of detecting vehicles with different scales.

Ammour et al. [28] used a two-stage network to solve the vehicle detection problem. In the first stage, the candidate regions were extracted, and a mean-shift algorithm was used to segment the images. In the second stage, vehicles would be detected here. The VGG16 model was applied to extract the region feature, and then a support vector machine (SVM) classifier was used. With the extracted features, the classifier could judge whether the object was a vehicle or a nonvehicle. Due to the combined phases that the model used (mean-shift segmentation [29], VGG16 [30] feature extraction, SVM classification), the amount of computation load was large, and object localization performance was excellent. It was a little similar to the series of R-CNN approaches where the algorithm suggested adding a region proposal algorithm for the rough object localization [31].

Reinforcing the vehicle feature extraction from convolutional neural networks is also a way to improve vehicle detection performance. Zhu et al. [32] designed a global relational block to enhance the fusion of features. With this structure, they improved the ability to detect small targets without reducing the speed of model detection. Liu et al. [33] proposed a new vehicle detection strategy to achieve each vehicle-rotated bounding box. Based on the new representation of vehicle targets, they transformed the vehicle detection task into a

saliency object detection task. Shao et al. [34] used the prior knowledge of coastline and made an improvement on YOLOv5 to adaptively obtain better feature fusion.

Different from existing works, we propose to alleviate the mismatch between discretely changed receptive fields and continuously changed vehicle scales via a Parallel RPN, which can generate finer proposals covering more objects. Then, we propose the Scale-NMS that sets different thresholds for proposals from different FPN layers. In addition, we improve the detection of densely arranged vehicles by the Density-assigner which performs label assignment in a dividing and conquering manner.

## 3. Method

### 3.1. Overall Detection Framework

In this paper, a specific vehicle detection framework for aerial images is proposed. Firstly, anchors are generated on different FPN layers and set position by Parallel RPN. Furthermore, these anchors are screened out to a fixed number in scale-based NMS and form a proposal list. The proposals are then assigned to ground truth in the R-CNN stage by a density-based assigning strategy. Figure 1 shows the architecture of the proposed vehicle detection framework. A detailed description and the motivation for the proposed work are given in the following sections.

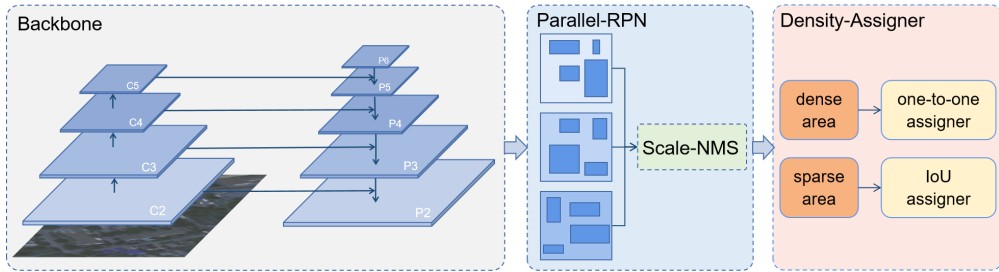

**Figure 1.** The framework of our proposed vehicle detection network. The network takes ResNet with FPN as the backbone and consists of a Parallel RPN head. The Parallel RPN head predicts foreground scores and refined anchor boxes. The scale-based NMS strategy is used to filter superfluous proposals according to different scales. Then, the proposals are sent to the R-CNN stage for further detection.

### 3.2. Parallel RPN

A parallel RPN structure is designed to make use of convolutional kernels with different receptive fields: $3 \times 3$, $5 \times 5$, and $7 \times 7$. Though we can obtain feature maps with different scales in FPN, the raw anchors generated with manually set scales and ratios have some limitations. For example, if the original image is $800 \times 800$ pixels, when using $3 \times 3$ convolutional kernels to obtain $200 \times 200$ and $100 \times 100$ feature maps, we can obtain theoretical receptive fields close to $12 \times 12$ and $24 \times 24$, respectively. Otherwise, if we use $5 \times 5$ and $7 \times 7$ convolutional kernels in $200 \times 200$ feature maps, the theoretical receptive fields are close to $20 \times 20$ and $28 \times 28$. As a result, convolutional kernels with different sizes can make up for the noncontinuity of receptive fields and improve the quality of proposals. The architecture of Parallel RPN is shown in Figure 2.

Given that the feature map sizes of different FPN layers vary by multiple, the deduced receptive field of different layers changes discretely, leading to the large gap between the receptive field size between adjacent FPN layers. However, the vehicle size approximates a continuous change in the aerial imagery; therefore, the discretely changed receptive field is not fine enough to handle the vehicle size variance. As a result, the choice of proposals from different convolutional kernels should also be adjusted dynamically according to the specific situation. In the traditional structure of RPN, proposals are generated together with classification scores (confidence). However, once the final proposals are chosen by their rankings, the scores will never be used again.

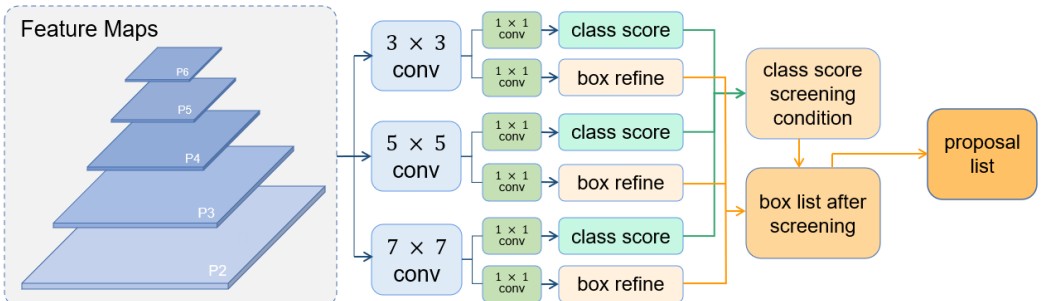

**Figure 2.** The structure of our proposed Parallel RPN. It consists of three convolutional branches with $3 \times 3$, $5 \times 5$, and $7 \times 7$ kernel sizes after FPN output. Three groups of bounding box refinements are screened by classification scores and finally compose a proposal list.

In our design, the exact scores are used to screen proposals from different convolutional kernels. The detailed equation and explanation of the architecture are given as follows.

$$B_3 = \frac{\overline{C_3}}{\overline{C_3} + \overline{C_5} + \overline{C_7}} \times N$$

$$B_5 = \frac{\overline{C_5}}{\overline{C_3} + \overline{C_5} + \overline{C_7}} \times N \qquad (1)$$

$$B_7 = \frac{\overline{C_7}}{\overline{C_3} + \overline{C_5} + \overline{C_7}} \times N$$

where $B_i$ means the number of proposals selected from the $i \times i$ convolutional layer, $\overline{C_i}$ means the average score of generated proposals from the $i \times i$ convolutional layer, and $N$ is the final number of proposals selected to construct the proposal list.

The proposed Parallel RPN consists of three parallel convolutional layers with kernel sizes of $3 \times 3$, $5 \times 5$, and $7 \times 7$. After convolution, we can obtain three groups of foreground classification scores and corresponding box refinements. Then, to screen out proper anchor boxes, we take the average classification score of each group as allocation criteria. The ratio of average scores among the three is that of proposals screened for the three groups. To avoid any branch being out of action, a minimum number of $0.2N$ for each branch is set to ensure that the proposals are screened from all three branches.

### 3.3. Density-Assigner

In terms of vehicle objects in aerial images, they are generally of small size. On the other hand, a considerable number of vehicles are densely arranged, such as being distributed along the road and gathering in parking lots. For vehicle detection datasets with horizontal annotations, this can cause many overlapping bounding boxes in areas of dense distribution. Meanwhile, NMS is a strategy to eliminate superfluous anchor boxes generated from RPN. To those vehicle objects that not only have tiny sizes but also distribute densely, proposals outputs from the RPN stage are also tiny and dense. In this case, it is very hard for NMS to judge which predicted box accurately locates the vehicle. In dense areas, proposals generated may be insufficient, which means that some foregrounds are not covered by RPN's proposals. At the same time, some proposals may contain more than one object and have high classification scores. Using a strict NMS strategy in this situation is not optimal, since it can cause a lack of appropriate proposals.

Thus, we designed a scale-based NMS strategy. The NMS threshold is selected according to the FPN layer where proposals come from. Proposals from the deeper FPN layer will pass through a more strict NMS. As a result, more proposals of small sizes in dense areas from large feature maps are kept for use in the following R-CNN stage.

After distinguishing those proposals from their FPN output layers, different NMS thresholds are applied according to the layers, respectively. Generally, the deeper the FPN

layer is, the larger the average size of proposals generated. To set the NMS threshold by increasing the deepening of FPN layers, we are able to keep more proposals for dense and small sizes.

Generally, vehicles are most commonly seen along roads, in parking lots, or in individual residential areas. Due to a large number of vehicles, they often cause severe traffic jams or crowded parking lots. Therefore, the distribution of vehicle objects tends to be dense in a certain area. Considering that most vehicle detection datasets adopt horizontal bounding boxes as their labeling mode, an overlapping box in dense traffic regions is a nonnegligible issue. The overlapping boxes of vehicles are shown in Figure 3. To improve the detection performance on these densely arranged objects, we propose to take the prior information into consideration to assist network learning. In other words, we propose to use the vehicles' density information to assist the label assignment and design a Density-assigner, as shown in Figure 4.

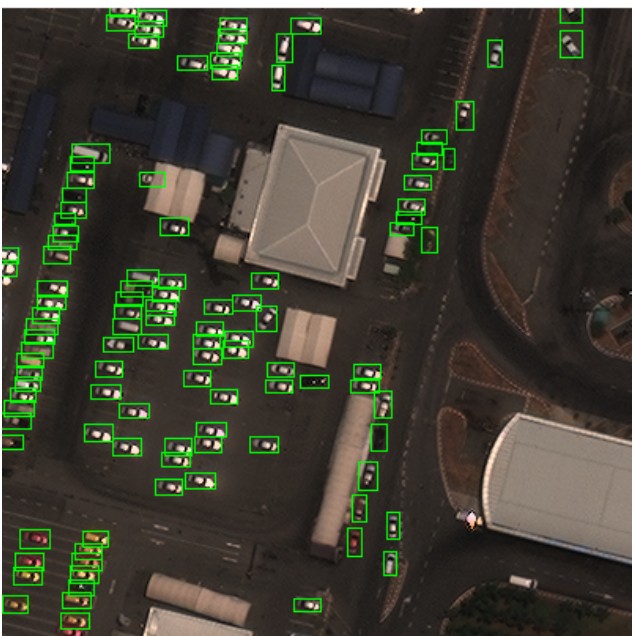

**Figure 3.** The overlapping boxes of vehicles are crowded at a scene of a parking lot in the xView-vehicle dataset. When the vehicles are not horizontally arranged or vertically arranged in the strict sense, overlaps will appear in dense areas such as parking lots and urban roads.

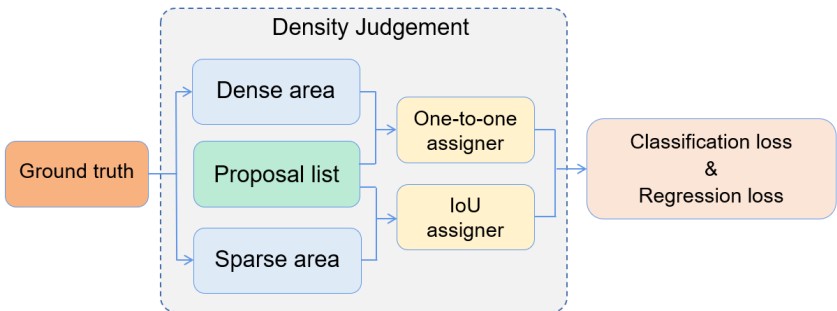

**Figure 4.** The structure of our proposed Density-assigner. After the region proposal network, a large number of proposals are generated. Then, in the R-CNN stage, these proposals are assigned as positive samples or negative samples according to the density judgment. If a ground truth is in a dense area, it will be assigned to only a single positive sample. After the assignment, samples are used to calculate classification loss and regression loss in the training process as usual.

For a vehicle that parks in the open countryside, proposals overlapping within a proper IoU threshold around it can all be used as a positive guide to supervising the bounding box regression training process. However, when the vehicle ground truths are dense in a region, not all proposals around the vehicle object with corresponding IoU are suitable for the bounding box regression training process. Those proposals between two ground truths are more likely to be assigned as positive samples to both two ground truths. Under this circumstance, the selected positive samples are of poor quality, and they may cause confusion in the bounding box regression training process. The visualization of easily-confused proposals is shown in Figure 5.

Inspired by the ideology of one-to-one detection in the transformer-based method DETR [35], we proposed a density-based assigner. This assigner will first judge whether the ground truths are in a dense area, and then conduct a one-to-one positive sample assignment. Since many proposals in dense areas were obtained, it is necessary to assign positive samples in a more strict way, eliminating redundant false predictions that cannot be deleted by NMS. The function of RPN is to give rough proposals for all the foreground objects, while that of R-CNN predicts the finally precise classes and locations. For those objects in dense areas, more proposals from RPN are reserved. As a result, any ground truth in these areas will be assigned to only a single proposal that has accordingly maximum IoU in the R-CNN stage.

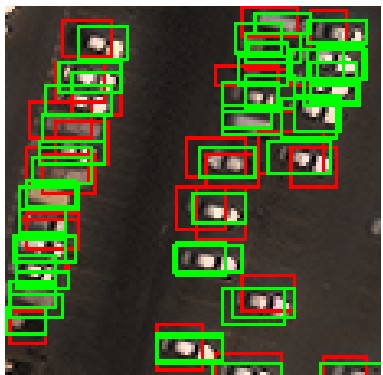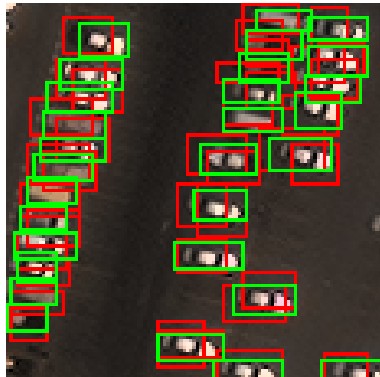

**Figure 5.** The green bounding boxes are positive samples that have been assigned to vehicle ground truths, while red bounding boxes are negative samples that will not be trained in the bounding box regression training process. The left picture shows the positive samples assigned only by calculating IoU. In this case, we choose 0.7 as the IoU threshold. Every proposal which has IoU greater than the threshold is assigned to the ground truth with max IoU. The right picture shows the positive samples assigned by the Density-assigner. In the dense area, only a single proposal is assigned to each ground truth as the positive sample. By contrast, positive samples assigned by the Density-assigner are of high quality, and poor proposal samples, which locate among several ground truths, are no longer assigned in this case.

Now that we choose the one-to-one assigning strategy in vehicle-dense areas, the definition of dense area is also a critical point worth studying. Firstly, and intuitively, any vehicle object ground truth which has IoU with another vehicle object ground truth belongs to the dense area, obviously. However, if vehicles lie orderly along a road or in a parking lot by coincidence, the ground truths of these vehicle objects may have no overlaps. Under this circumstance, to group the ground truths of vehicle objects together with dense vehicle objects, we design a new formula to measure whether a vehicle object belongs to objects in a dense area, as is shown in the following equations.

$$D_O^2 = (w_a - w_b)^2 + (h_a - h_b)^2 \tag{2}$$

where $w_a$ represents the width of ground truth bounding box A, and $w_b$ represents the width of ground truth bounding box B. Similarly, $h_a$ represents the height of ground truth

bounding box A, and $h_b$ represents the height of ground truth bounding box B. In this equation, $D_O^2$ is used to measure the difference of bounding box outlines. If the outlines of two bounding boxes are obviously different, they will be less likely to cause confusion in label assignments. In this situation, the low value of IoU is able to eliminate samples of poor quality.

$$D_C^2 = \frac{(cx_a - cx_b)^2 + (cy_a - cy_b)^2}{(\frac{w_a + w_b}{2})^2 + (\frac{h_a + h_b}{2})^2} \tag{3}$$

where $cx_a$ represents the center point's x-coordinate of ground truth bounding box A, and $cx_b$ represents the center point's x-coordinate of ground truth bounding box B. Similarly, $cy_a$ represents the center point's y-coordinate of ground truth bounding box A, and $cy_b$ represents the center point's y-coordinate of ground truth bounding box B. $D_C^2$ here measures the adjacent level of two ground truth center points. The closer they are to each other, the more likely it is that they will cause confusion in label assignments.

$$I_D = exp\left(-\frac{\sqrt{D_O^2 + D_C^2}}{C}\right) \tag{4}$$

where $C$ is a constant that is closely related to the dataset to constrain the range. With the help of hyperparameter $C$, we can control the sensitivity, which makes it more convenient to distinguish the dense area and the sparse area. In this equation, the final value is mapped to the range of 0 to 1 by exponential form. In our definition, the value $I_D$ can measure the dense level of every two ground truth bounding boxes. If the value $I_D$ of two ground truth bounding boxes exceeds the density threshold, these two ground truth bounding boxes are judged to be in the dense area.

Placing the Density-assigner together with Scale-NMS, they form a structure that first releases and then contracts. Only when Scale-NMS provides a sufficient quantity of proposals can the Density-assigner then carry out the one-to-one assignment in a dense area. The Density-assigner makes sure that every ground truth is assigned to the optimal positive sample as far as possible and eliminates the phenomenon of missing assignment.

## 4. Results

### *4.1. Dataset*

To evaluate the effectiveness of our proposed method, we chose two datasets that include a large number of vehicle objects: (1) the AI-TOD dataset [36], a dataset with the smallest vehicle object size in the Earth observation community, and (2) the xView dataset [37], a very complex dataset generated from the Earth observation satellite which is mainly concerned with different kinds of vehicles. To study the behavior of our proposed method, we carried out some processing on these two datasets before experimenting. The details will be shown in the following.

#### 4.1.1. AI-TOD-Vehicle Dataset

The images in the AI-TOD dataset came from two imaging sources; one is the Earth observation satellite and the other is UAV. The original AI-TOD dataset includes 28,036 images with 700,621 objects of eight classes annotated with horizontal bounding boxes (HBBs). The images from both sources were originally 800 × 800 pixels in size. In our task, we only detect vehicle objects, so we screened out all the images with vehicle objects and redistributed the training set and validation set with a rate of nearly 4:1. The absolute object size of vehicles is only 12.6 pixels, and the size range of different vehicles is from 8 pixels to 20 pixels. Table 1 shows the characteristics of the AI-TOD-vehicle dataset.

**Table 1.** Characteristics of the AI-TOD-vehicle dataset.

| Sets | Characteristics | Values |
|---|---|---|
| Train | Number of images | 5749 |
| | Number of bounding boxes | 248,042 |
| | Average vehicles per picture | 43 |
| Validation | Number of images | 1443 |
| | Number of bounding boxes | 59,904 |
| | Average vehicles per picture | 41 |
| Total | Number of images | 7192 |
| | Number of bounding boxes | 307,946 |
| | Average vehicles per picture | 43 |

### 4.1.2. xView-Vehicle Dataset

The original data of xView were collected from WorldView-3 satellites at an average of 0.3 m ground sample distance. It provided higher-resolution imagery than most public satellite imagery datasets. In addition, xView is one of the largest and most diverse publicly available object detection datasets to date, with over 1 million objects across 60 classes in over 1400 km$^2$ of imagery. The images in the dataset were originally $400 \times 400$ pixels in size. After checking the 60 classes carefully, we eventually chose 31 of them which were related to vehicles, including various kinds of passenger vehicles, trucks, railway vehicles, and engineering vehicles. Considering that our proposed method is focused on tackling the scale variance and dense arrangement issues, we integrated these 31 classes of vehicles into a single class entitled "vehicle". In this way, the experiments can show our progress in the bounding box regression training process, which leads to vehicle localization. The mean absolute object size of vehicles is only 14.1 pixels, and the size range of different vehicles is from 8 pixels to 33 pixels. Table 2 shows the characteristics of the xView-vehicle dataset.

**Table 2.** Characteristics of the xView-vehicle dataset.

| Sets | Characteristics | Values |
|---|---|---|
| Train | Number of images | 26,190 |
| | Number of bounding boxes | 371,534 |
| | Average vehicles per picture | 14 |
| Validation | Number of images | 5274 |
| | Number of bounding boxes | 81,091 |
| | Average vehicles per picture | 15 |
| Total | Number of images | 31,464 |
| | Number of bounding boxes | 452,625 |
| | Average vehicles per picture | 14 |

In this section, we assess the proposed method across the AI-TOD-vehicle and xView-vehicle datasets. Overall, the proposed method shows significant training results in terms of conventional metrics such as bbox_$mAP$, bbox_$AP_{0.5}$, and bbox_$AP_{0.75}$.

Firstly, we present our experiment settings, including training parameter settings and evaluation metrics. Then, we provide a benchmark of our proposed method together with different backbones and detector heads. Finally, the effectiveness of our proposed method is validated by extensive comparative experiments and ablation studies regarding different kinds of combinations.

### 4.2. Experiment Settings

#### 4.2.1. Parameter Settings

All the experiments were conducted on a computer with one single NVIDIA RTX 3090 GPU, and all the models training were built on PyTorch deep learning framework. We

took the ImageNet pretrained ResNet-50 together with FPN as the backbone. In specified situations, we used other backbones suitable for the detector heads.

In experiments of the AI-TOD-vehicle dataset, we used the stochastic gradient descent (SGD) optimizer for 12 epochs with 0.9 momenta, 0.0001 weight decay, and the batch size was 2. The initial learning rate was 0.01 and it decayed at epochs 8 and 11. In addition, in linear step warmup, warmup iterations were set to 2000 and the ratio was set to 0.001. In addition, the batch sizes of RPN and Faster R-CNN were set to 256 and 512, respectively. The number of proposals generated by the RPN head was set to 1000 for both RPN and our proposed Parallel RPN. The IoU threshold of defining positive training samples is set to 0.7 in the IoU assigner. The above training and inference parameters are used in all experiments, and other parameters not mentioned here will be given in specified situations.

In experiments of the xView-vehicle dataset, we used the stochastic gradient descent (SGD) optimizer for 12 epochs with 0.9 momenta, 0.0001 weight decay, and the batch size was 4. The initial learning rate was 0.01 and it decayed at epochs 8 and 11. In addition, in linear step warmup, warmup iterations were set to 10,000 and the ratio was set to 0.001. The batch size of RPN and Faster R-CNN are both set to 256. The number of proposals generated by the RPN head was set to 1000 for both RPN and our proposed Parallel RPN. The IoU threshold was set to 0.7 in the IoU assigner. The above training and inference parameters were used in all experiments, and other parameters not mentioned here will be given in specified situations.

### 4.2.2. Evaluation Metric

AP (average precision) is the main metric that is used to evaluate the performance of our proposed method. Specifically, $AP_{0.5}$ means that the IoU threshold of defining true positive predictions is 0.5, and, similarly, $AP_{0.75}$ means that the IoU threshold of defining true positive predictions is 0.75, which is more strict. Considering that vehicle objects in the datasets are of various sizes, we utilize $AP_s$ and $AP_m$ to evaluate the performance of detecting vehicles at small and medium scales. If a vehicle object is smaller than $32 \times 32$ pixels, it belongs to a small scale, and other vehicles in $32 \times 32$ pixels to $96 \times 96$ pixels, such as various kinds of engineering vehicles, belong to the medium scale.

### 4.3. Benchmark

We conducted many experiments over different baseline detectors, including both anchor-based and anchor-free detectors. Considering that most of the vehicle objects in aerial images are of small scale, the performance of these baselines is much worse than that of the general scenes such as PASCAL VOC [38,39] and MS COCO [40]. To handle this problem, we added NWD together with Faster R-CNN and DetectoRS as new baselines, which are more suitable for vehicle detection in aerial images. Table 3 shows the performance of our proposed method, which was trained for 12 epochs on the AI-TOD dataset and compared to other models. In this table, we can see that our proposed method made progress on the basis of tiny object detection baselines. Except for the small size of vehicle objects, we grasped another two characteristics, uniform change of size and dense distribution in specific areas, and our proposed method was designed according to these. Together with the NWD method, our proposed method further improves the mAP of vehicle object detection from 22.9% to 24.2% on the basis of Faster R-CNN.

To further verify our proposed method, we conducted another group of experiments on the xView-vehicle dataset, and the results are shown in Table 4. Since there are more kinds of vehicles in the xView-vehicle dataset, their sizes change in a larger range and the number of vehicle objects is a lot higher, which means this dataset is suitable to evaluate the proposed method. Compared with the AI-TOD-vehicle dataset, the images are smaller in size, and the average vehicle object number per image is lower. As a result, the performance of baselines is a little better than that in the AI-TOD-vehicle dataset. Together with the NWD method, our proposed method further improved the mAP of vehicle object detection from 22.9% to 24.2% on the basis of Faster R-CNN.

**Table 3.** Performance of our proposed method, trained for 12 epochs on the AI-TOD-vehicle dataset, and compared to other models. Bold numbers are the best results and underlined numbers are the second best.

| Method | Backbone | *mAP* (%) | $AP_{0.5}$ (%) | $AP_{0.75}$ (%) | $AP_s$ (%) | $AP_m$ (%) |
|---|---|---|---|---|---|---|
| FCOS [41] | ResNet-50-FPN | 12.0 | 25.5 | 9.8 | 10.3 | 33.7 |
| YOLOv4 [42] | DarkNet-53 | 10.9 | 25.2 | 8.4 | 8.9 | 33.0 |
| RetinaNet [43] | ResNet-50-FPN | 10.0 | 29.8 | 4.1 | 9.5 | 36.3 |
| ATSS [20] | ResNet-50-FPN | 12.2 | 25.3 | 10.1 | 10.3 | 33.4 |
| Faster R-CNN [4] | ResNet-50-FPN | 12.8 | 30.9 | 7.8 | 12.2 | 38.4 |
| Cascade R-CNN [5] | ResNet-50-FPN | 13.2 | 31.2 | 8.0 | 12.3 | 38.6 |
| Dynamic R-CNN [44] | ResNet-50-FPN | 10.1 | 23.8 | 6.5 | 10.2 | 40.0 |
| Tridentnet [45] | ResNet-50-FPN | 12.4 | 33.1 | 6.4 | 12.2 | 43.1 |
| Cascade RPN [46] | ResNet-50-FPN | 16.3 | 47.6 | 12.2 | 18.7 | 38.1 |
| DetectoRS [47] | ResNet-50-FPN | 15.6 | 44.3 | 5.8 | 15.5 | 43.2 |
| Faster R-CNN + NWD [1] | ResNet-50-FPN | 22.9 | 58.7 | 10.9 | 22.4 | 43.9 |
| DetectoRS + NWD [1] | ResNet-50-FPN | 23.7 | <u>59.1</u> | 11.4 | 23.1 | <u>44.2</u> |
| Faster R-CNN + NWD [1] + ours | ResNet-50-FPN | <u>24.2</u> | 59.0 | **12.9** | <u>23.7</u> | 43.0 |
| DetectoRS + NWD [1] + ours | ResNet-50-FPN | **25.1** | **61.2** | <u>12.7</u> | **24.9** | **45.9** |

[1] NWD refers to [23] , which is an assigning strategy in the RPN stage aimed at detecting tiny objects.

**Table 4.** Performance of our proposed method, trained for 12 epochs on the xView-vehicle dataset, and compared to other models. Bold numbers are the best results and underlined numbers are the second best.

| Method | Backbone | *mAP* (%) | $AP_{0.5}$ (%) | $AP_{0.75}$ (%) | $AP_s$ (%) | $AP_m$ (%) |
|---|---|---|---|---|---|---|
| FCOS [41] | ResNet-50-FPN | 12.9 | 33.7 | 6.4 | 12.8 | 19.8 |
| YOLOv4 [42] | DarkNet-53 | 11.8 | 30.3 | 3.9 | 10.8 | 19.9 |
| RetinaNet [43] | ResNet-50-FPN | 12.7 | 32.7 | 6.7 | 12.7 | 18.8 |
| ATSS [20] | ResNet-50-FPN | 14.2 | 34.9 | 8.4 | 14.3 | 21.3 |
| Faster R-CNN [4] | ResNet-50-FPN | 14.3 | 42.2 | 4.3 | 14.3 | 11.7 |
| Cascade R-CNN [5] | ResNet-50-FPN | 15.4 | 44.3 | 5.4 | 15.4 | 14.9 |
| Dynamic R-CNN [44] | ResNet-50-FPN | 13.9 | 35.7 | 7.0 | 13.8 | 22.1 |
| Tridentnet [45] | ResNet-50-FPN | 16.3 | 42.6 | 6.9 | 15.7 | 23.1 |
| Cascade RPN [46] | ResNet-50-FPN | 17.8 | 45.3 | 7.2 | 16.6 | 23.8 |
| DetectoRS [47] | ResNet-50-FPN | 18.2 | 49.2 | 7.6 | 18.1 | 23.6 |
| Faster R-CNN + NWD | ResNet-50-FPN | 24.6 | 66.7 | 11.1 | 24.2 | 34.6 |
| DetectoRS + NWD | ResNet-50-FPN | 25.3 | <u>67.2</u> | 11.4 | 24.9 | 32.9 |
| Faster R-CNN + NWD + ours | ResNet-50-FPN | <u>25.8</u> | 67.0 | <u>12.5</u> | <u>25.4</u> | <u>35.7</u> |
| DetectoRS + NWD + ours | ResNet-50-FPN | **26.3** | **67.9** | **13.7** | **28.0** | **37.7** |

Improvements based on the strong baseline verify the effectiveness of our proposed method. Our proposed method originates from the prior knowledge of vehicles, continuous change of size, and dense distribution of vehicles in specific areas. Combined with the method of tiny object detection NWD, we further improved the performance both on the AI-TOD-vehicle dataset and the xView-vehicle dataset. Vehicle detection in aerial images still remains a tough problem, and determining how to make full use of the prior knowledge is one of the key points.

### 4.4. Ablation Study

In this section, we show the performance of the different modules in our proposed method, respectively, and extra experiments of the ablation study will be added to further enrich our explanation.

4.4.1. Combination of Parallel RPN

Taking the Parallel RPN with three branches as an example, the visualization of proposals generated by it is shown in Figure 6. In this image, we can see that some of the vehicle objects are detected by all three RPN branches, and then similar proposals are generated, respectively. However, there are still some vehicle objects which can only be detected by a single RPN branch or two branches, and proposals seem to be generated from various combinations of different RPN branches and different FPN layers. As a result, different RPN branches are complementary together with their source of different FPN layers. When the size of vehicle objects changes uniformly, convolutional kernels in Parallel RPN with diverse receptive fields can suit the remedy to the case. The proposals are generated complementarily, and this will ease the problem of missing detection to some extent.

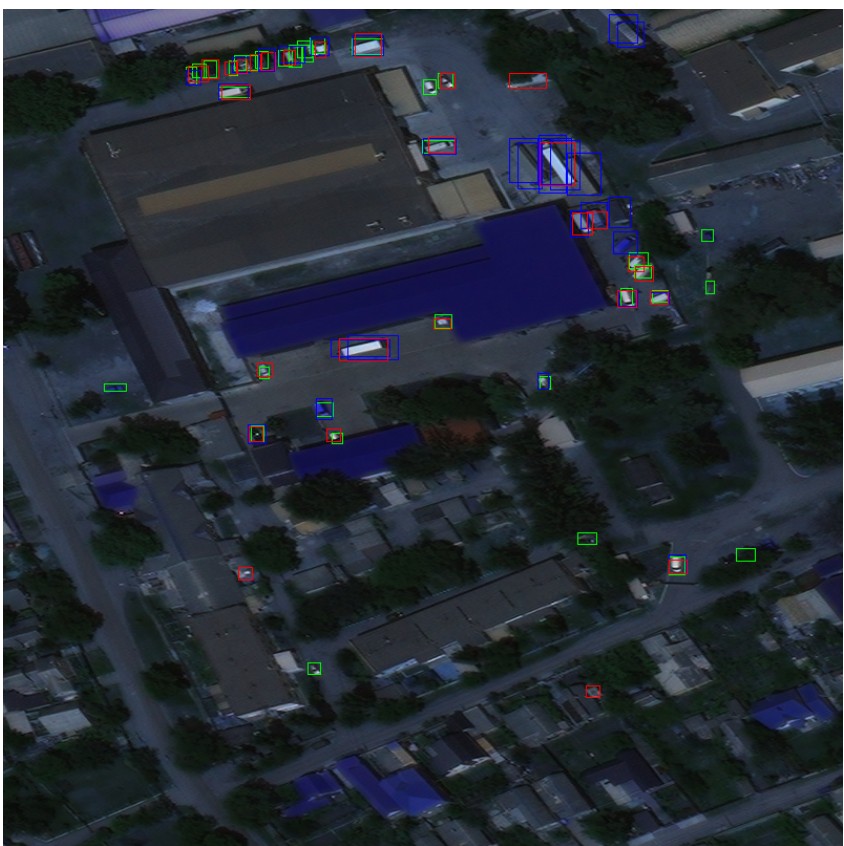

**Figure 6.** The visualization of proposals generated by the three branches of Parallel RPN in the AI-TOD-vehicle dataset. In this image, proposals in green, blue, and red are generated from the $3 \times 3$ branch, $5 \times 5$ branch, and $7 \times 7$ branch, respectively.

To further validate our proposed Parallel RPN module, we conducted several comparison experiments on the AI-TOD-vehicle dataset. Table 5 shows the comparison of Parallel RPN with various combinations of different branches. Here, $AR$ represents the average recall. In contrast to the original $3 \times 3$ RPN, $5 \times 5$ RPN performs better, and the combination of $3 \times 3$, $5 \times 5$, and $7 \times 7$ reaches even higher. To further verify the design of the Parallel RPN, we conducted another group of comparison experiments on the xView-vehicle dataset. The results are shown in Table 6. We can find that the metric $AR$ also increases a little, which gives expression to the relief of missing detection. The optimal combination of RPN branches in Parallel RPN is relevant to the task and the dataset. Considering the trade-off of performance and parameters, we suggestively choose the $3 \times 3$, $5 \times 5$, and $7 \times 7$ branches as our proposed Parallel RPN module.

**Table 5.** This is a comparison of Parallel RPN on the AI-TOD-vehicle dataset, which contains various combinations of different branches. Bold numbers are the best results.

| RPN Branch | $mAP$ (%) | $AP_{0.5}$ (%) | $AP_{0.75}$ (%) | $AR$ (%) |
|---|---|---|---|---|
| original $3 \times 3$ | 22.9 | 58.7 | 10.9 | 27.5 |
| single $5 \times 5$ | 23.6 | 58.6 | 11.3 | 27.6 |
| single $7 \times 7$ | 22.7 | 58.5 | 10.8 | 29.4 |
| $3 \times 3$ and $5 \times 5$ | 23.7 | 58.9 | 11.9 | 32.2 |
| $5 \times 5$ and $7 \times 7$ | 23.9 | 58.8 | 11.4 | 31.5 |
| $3 \times 3, 5 \times 5$ and $7 \times 7$ | **24.2** | **59.0** | **12.9** | **33.0** |

**Table 6.** This is a comparison of Parallel RPN on the xView-vehicle dataset, which contains various combinations of different branches. Bold numbers are the best results.

| RPN Branch | $mAP$ (%) | $AP_{0.5}$ (%) | $AP_{0.75}$ (%) | $AR$ (%) |
|---|---|---|---|---|
| original $3 \times 3$ | 24.6 | 66.7 | 11.1 | 35.7 |
| single $5 \times 5$ | 25.2 | 66.6 | 11.5 | 36.1 |
| single $7 \times 7$ | 24.9 | 66.7 | 11.6 | 34.5 |
| $3 \times 3$ and $5 \times 5$ | 25.6 | 66.8 | 12.1 | 36.3 |
| $5 \times 5$ and $7 \times 7$ | 25.8 | **67.1** | 12.1 | 36.4 |
| $3 \times 3, 5 \times 5$ and $7 \times 7$ | **25.8** | 67.0 | **12.5** | **37.1** |

### 4.4.2. Design in Scale-NMS

Proposals are generated from the RPN head according to different FPN output layers and convolutional layers. Before being gathered into the final proposal list, NMS is applied to obtain rid of repetitive proposals generated around the same vehicle object. Traditional NMS uses the same threshold and equally treats proposals from different FPN layers, for example, 0.7. Once the proposal with the max score is settled, all the other lower score proposals with IoU higher than 0.7 around it will be eliminated. Notice that feature maps from different FPN output layers are of respective scales; we can change thresholds to screen out proposals from different FPN output layers.

Table 7 shows the exploration of different NMS thresholds selected according to FPN output layers. Generally, vehicle objects of smaller size come from lower FPN layers. Since we want to keep more proposals with a small scale, the NMS threshold from the bottom FPN layer is set lower. In the experiments, we change the NMS threshold uniformly from top to bottom, and Figure 7 shows the visualization of proposals after traditional NMS and our proposed Scale-NMS. We can see that our proposed Scale-NMS can keep more proposals for the following Density-assigner.

**Table 7.** The results under different NMS thresholds in the AI-TOD-vehicle dataset. Bold numbers are the best results.

| NMS Thresholds | $mAP$ (%) | $AP_{0.5}$ (%) |
|---|---|---|
| 0.7 to 0.9 [1] | 24.1 | 58.9 |
| 0.7 to 0.85 | **24.2** | **59.0** |
| 0.7 to 0.8 | **24.2** | 58.9 |
| 0.7 to 0.75 | 23.9 | 58.7 |

[1] The small number is the NMS threshold from the upper FPN output layer, while the large number is that from the bottom FPN output layer. The middle layers have uniformly distributed thresholds in this range.

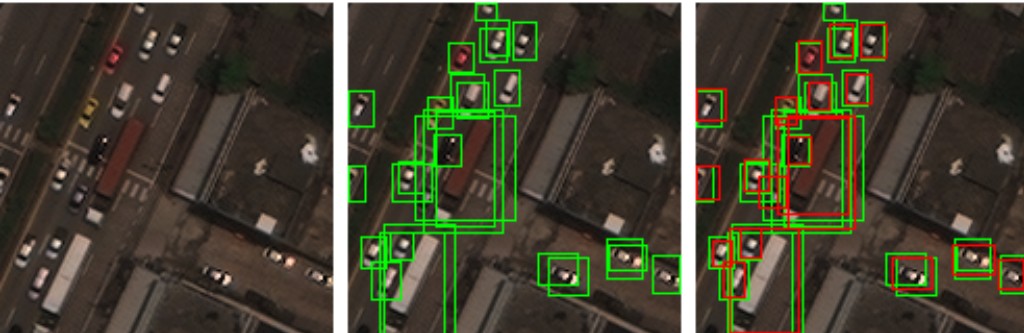

**Figure 7.** The visualization of proposals kept after traditional NMS and our proposed Scale-NMS. The left image is the original aerial image, the middle one is the result of traditional NMS with a threshold of 0.7, and the right one is the result from our proposed method with an NMS threshold from 0.7 to 0.85. The red bounding boxes are proposals kept in our method, which are supplements to the proposal list.

### 4.4.3. Hyperparameter in Density-Assigner

The constant C in Equation (4) and the density threshold $I_{thr}$ are the two hyperparameters. When designing our proposed Density-assigner, the constant C was used to constrain the range to prevent the value $I_D$ from being too small. As a result, we empirically set C to 12 in the AI-TOD-vehicle dataset and 16 in the xView-vehicle dataset, according to [23]. Then, we conducted a group of experiments to settle the threshold. The results are shown in Table 8. Additionally, we counted the number of vehicles that are supposed to locate in the dense area upon the optimal results. There are 66,893 instances in the AI-TOD-vehicle dataset, and 87,745 in the xView-vehicle dataset.

**Table 8.** The results under different density thresholds in the AI-TOD-vehicle dataset and the xView-vehicle dataset. Bold numbers are the best results.

| Dataets | Density Thresholds | $mAP$ (%) | $AP_{0.5}$ (%) |
|---|---|---|---|
| AI-TOD-vehicle | 0.2 | 23.9 | 58.7 |
| | 0.25 | **24.2** | **59.0** |
| | 0.3 | 24.1 | 58.7 |
| | 0.35 | 24.1 | 58.9 |
| xView-vehicle | 0.2 | 25.7 | 66.8 |
| | 0.25 | 25.6 | 66.9 |
| | 0.3 | **25.8** | **67.0** |
| | 0.35 | 25.8 | 66.9 |

### 4.4.4. Individual Effectiveness of Each Module

At the end of the ablation study section, we verify the modules of our proposed method in the AI-TOD-vehicle dataset. The baseline detector is a Faster R-CNN + NWD detector, and our proposed method is separated into two modules, Parallel RPN and Density-assigner. Density-assigner is the combination of our proposed scale-based NMS strategy and density-based assigner, since the expansion and contraction structure is coupled. Results are shown in Table 9. Here, we use the high-precision evaluation metric $AP_{0.75}$ to demonstrate our improvement in vehicle detection, especially vehicle localization.

**Table 9.** Ablation of individual effectiveness of Parallel RPN and Density-assigner.

| Parallel RPN | Density-Assigner | $mAP$ (%) | $AP_{0.75}$ (%) |
|:---:|:---:|:---:|:---:|
|  |  | 22.9 | 10.9 |
| ✓ |  | 23.5 | 11.6 |
|  | ✓ | 23.4 | 11.4 |
| ✓ | ✓ | 24.2 | 12.9 |

*4.5. Visualization*

The visualization results of our proposed method and comparison with the Faster R-CNN + NWD detector are shown in Figure 8. From the results, we can see that our proposed method made progress in vehicle detection, especially in vehicle localization. Furthermore, in some dense areas, our proposed method performs well and has a lower missed detection rate. It indicates that when equipped with our method, the anchor-based detector seems capable of handling vehicles in a dense area with overlapped ground truth bounding box. This is a successful use of prior knowledge in vehicle detection.

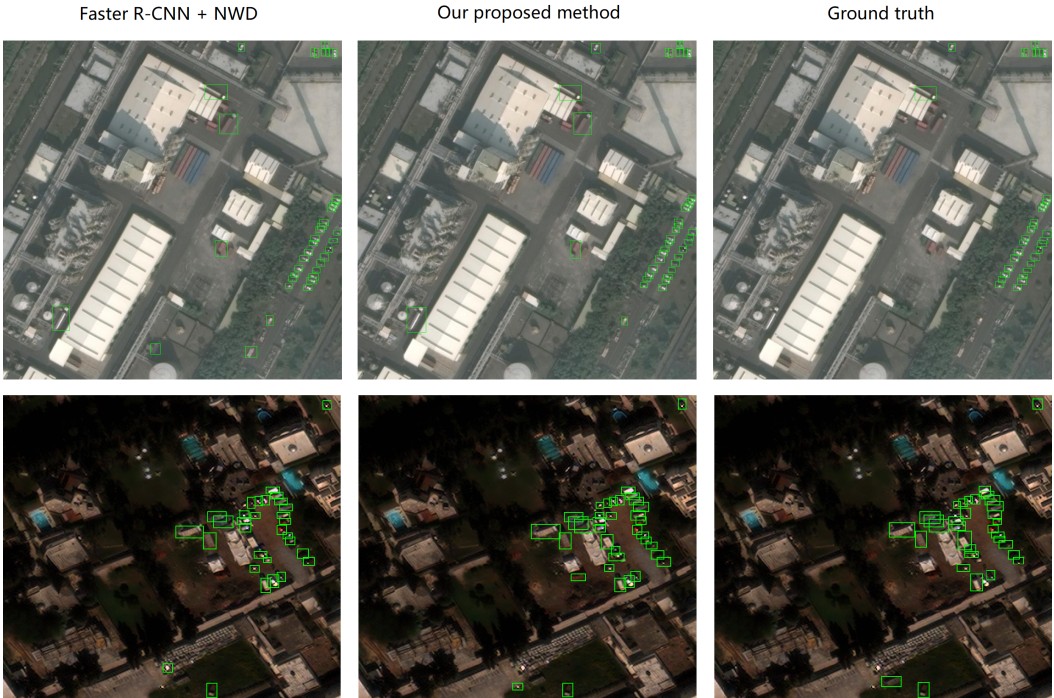

**Figure 8.** The visualization of vehicle detection results. The first group of images shows prediction results and ground truths in the AI-TOD-vehicle dataset, while the other group shows that of the xView-vehicle dataset. Note that our proposed method is applied to the Faster R-CNN + NWD method.

## 5. Discussion

Overall, extensive experiments and analyses demonstrate the effectiveness of our proposed method. By demonstrating the anchor-based detector's capability to handle vehicles in a dense area with overlapped ground truth bounding box when equipped with our method, we showcase a successful use of prior knowledge in vehicle detection. Through careful observation of the scale variation and distribution pattern of vehicles in aerial images, we identified an inductive bias that can guide network training and improve its performance.

From the results in Figure 8, we can find some typical failure cases of the proposed method. They are mainly false positive (FP) samples. Some of them are caused by the label noise phenomenon in the dataset itself, but most of them are real false positive samples.

These FP samples are confusable and may even be detected improperly by human beings. From the feature level, false positive samples are similar to real vehicle objects. To handle this problem, context information should be used more widely. This will be further explored in our future work.

## 6. Conclusions

In this paper, we proposed a new framework for detecting vehicles in aerial images. Specifically, we mainly addressed the scale variance and dense arrangement issues. First of all, we designed a Parallel RPN structure to relieve the mismatch between continuously changed vehicle scale and discrete feature receptive field. Then, we leveraged the prior knowledge of the object's density information and proposed the Scale-NMS and the Density-assigner strategy, enhancing detection in dense areas. Extensive experiments in two datasets show that our approach can improve the performance of vehicle detection in aerial images.

**Author Contributions:** Conceptualization, X.K. and Y.Z.; methodology, X.K. and C.X.; validation, X.K., S.T., and Y.Z.; investigation, X.K., Y.Z., S.T., and C.X.; resources, C.X., S.T., and W.Y.; writing—original draft preparation, X.K.; writing—review and editing, Y.Z. and W.Y. All authors have read and agreed to the published version of the manuscript.

**Funding:** The research was partially supported by the CETC key laboratory of aerospace information applications under Grant SKX222010021.

**Institutional Review Board Statement:** Not applicable.

**Informed Consent Statement:** Not applicable.

**Data Availability Statement:** The datasets will be available at https://github.com/WHUKong/Dataset.git.

**Conflicts of Interest:** The authors declare no conflicts of interest.

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
