# Peer review of "Vehicle Detection in High-Resolution Aerial Images with Parallel RPN and Density-Assigner"

_remotesensing, doi:10.3390/rs15061659_

Round 1

Reviewer 1 Report

This paper presents a novel CNN-based method to detect vehicles in aerial images. Their method proposes solutions to these three problems associated to this detection task: small objects, large object variance and densely populated areas in the image.

They experimented on two datasets, and proved the superiority of the proposed technique on this problem w.r.t. a wide range of state-of-the-art techniques.

The paper is well written and understandable, but can merit of a proof reading session. E.g. I find the choice of the word "existence" in the abstract not well chosen in the senstence "the existence of tiny objects".

My largest remark is that the claimed problems connected with aerial vehicle detection are not quantitatively demonstrated. In the dataset description prargraphs, I would expect numbers indicating the average pixel size of the cars in the image, the within-class variance of this pixel size, and the distribution of distances between cars in the image. With these numbers, you can prove that these problems occur with this detection task.

Earlier work on this task, based on single-shot detectors, is not referenced, e.g. Ophoff, T., et al. Vehicle and Vessel Detection on Satellite Imagery: A Comparative Study on Single-Shot Detectors. Remote Sensing, 12 (7), 2020.

Patel, Krishna, Chintan Bhatt, and Pier Luigi Mazzeo. "Deep learning-based automatic detection of ships: An experimental study using satellite images." Journal of imaging 8.7 (2022): 182.

Shao, Jiangnan, et al. "Vessel detection from nighttime remote sensing imagery based on deep learning." IEEE Journal of Selected Topics in Applied Earth Observations and Remote Sensing 14 (2021): 12536-12544. de

Carvalho, Osmar Luiz Ferreira, et al. "Bounding box-free instance segmentation using semi-supervised iterative learning for vehicle detection." IEEE Journal of Selected Topics in Applied Earth Observations and Remote Sensing 15 (2022): 3403-3420.

Reviewer 2 Report

This manuscript is interesting. However, a range of problems remain to be solved before considering publication.

Line 6. 'the receptive field in the current detector is not fine enough to handle the slight scale'. What does 'fine' actually mean?

Line 7-8. It not clear: the densely arranged vehicles will introduce ambiguous positive samples in label assignment and false predictions that cannot be deleted by NMS.

Line 9. What is ‘prior attribution knowledge’

Repetitions or inappropriate phrases/statements: post-processing process (Line 48), a normalized Normalized (Line 99)

Line 263. To examine the behavior of our proposed method?Do you  mean 'To evaluate the performance'?

Line 373. the problem of missing detection to some extent----Did you discuss missing detection in the results? Why not evaluate the performance using Recall?

A  range of STOA methods are compared with the proposed one. The advantage of the proposed method lies in the selection of proposals on different scales of features maps. Therefore, comparison with methods using multi-scale feature fusion is strongly recommended.

Reviewer 3 Report

This is an interesting paper, very well structured and written, proposing a two-stage framework for vehicle detection in aerial imagery. They design a parallel RPN that exploits convolution layers of different receptive fields to alleviate the scale variation problem. They propose a scale-based NMS to hierarchically filter redundant proposals from different levels of the feature pyramid. Constructing two vehicle detection datasets based on AI-TOD and xView datasets containing many small objects- Performing different experiments with these two datasets wur demonstrate the effectiveness of the proposed method.

Just a couple of minor issues:

I think IoU threshold should be properly defined.

I think it is better to display the figures after their reference in the text and not before.

Reviewer 4 Report

The paper proposes an interesting approach for vehicle detection in aerial images. However, some modifications are required to make the article sounder for readers.

In the paper, there are some strong claims that might not be entirely accurate. For example, the abstract says: "At present, state-of-the-art object detectors do not perform well on vehicle detection in aerial images."

Several abbreviations have been used in the paper without being defined, such as NMS, FPN, RPN.

As one of the challenges, it was mentioned in the paper that the scale of vehicles is continuously changing. However, the scale of vehicles follows traffic regulations, so they do not vary enormously.

The parameters in the Density-Assigner section, such as ID and C and their impact on the density detection have not been explained clearly.

Round 2

Reviewer 2 Report

My only concern lies in the language. I strongly recommend careful and thorough proofreading by an English native speaker.
